# Effects of Personal Hygiene Habits on Self-Efficacy for Preventing Infection, Infection-Preventing Hygiene Behaviors, and Product-Purchasing Behaviors

**Hyun Jung Yoo and Eugene Song ***

Department of Consumer Science, Chungbuk National University, Cheongju 28644, Korea; yoohj@cbnu.ac.kr
* Correspondence: eugenesong@chungbuk.ac.kr

**Abstract:** Since there is no cure for the COVID-19 pandemic yet, personal hygiene management is important for protecting oneself from the deadly virus. Personal hygiene management comes from personal hygiene habits. Thus, this study investigated the association between personal hygiene habits, consumers' infection-prevention behaviors, and the effects of social support on the latter. Data were collected using a self-administered questionnaire survey of 620 Korean adults. An online survey agency was used to conduct the questionnaire over eight days, from 18 May to 25 May 2020. Data were analyzed using structural equation modeling. The results were as follows. First, personal hygiene habits positively affected self-efficacy for infection prevention ($\beta = 0.123$, $p < 0.01$). Moreover, personal hygiene habits indirectly affected virus spread-prevention behaviors ($\beta = 0.457$, $p < 0.000$) and product-purchasing behaviors for infection prevention ($\beta = 0.146$, $p < 0.01$) through self-efficacy for infection prevention. Second, informational support for infection prevention increased self-efficacy influence for infection prevention on the virus spread prevention behaviors among the public (composite reliability: $-2.627$). Thus, continued education of the public is imperative to ensuring compliance with personal hygiene practices. Furthermore, timely dissemination of relevant information on infection-prevention practices through various media during an infection outbreak is critical.

**Keywords:** hygiene habits; informational support; self-efficacy; infection prevention behaviors; COVID-19



## 1. Introduction

In recent years, global public health has been threatened continuously. The severe acute respiratory syndrome, novel influenza, and Middle East respiratory syndrome outbreaks in 2003, 2009, and 2015, respectively, resulted in substantial social and financial losses [1]. The COVID-19 pandemic, which first occurred in 2020, is a devastating global disaster with tremendous repercussions, such as border closures and quarantine measures.

In particular, to prevent the transmission of viruses, such as the viral pathogen for COVID-19, the relevant actors (i.e., the global population) must make sacrifices (such as practicing social distancing and purchasing and wearing masks) for the public's good [2–4]. Several studies have emphasized the importance of public compliance with preventive practices as protective measures during the COVID-19 pandemic [5–11]. Therefore, in this study, we focused on identifying the factors that promoted infection prevention behaviors among the public during the spread of infection.

Several studies that explored habits explained that personal habits and intentions were variables that contributed significantly to explaining behaviors [12–16]. These studies stated that human behavior is automatic and habitual. The social cognitive theory states that human behavior is influenced by the individual's efficacy, which refers to the individual's confidence in engaging in a behavior [17,18]. These studies argued that daily habits and self-efficacy influenced human behavior.

Countries worldwide have implemented an array of support measures for the at-risk global population to prevent the spread of infection. To prevent the spread of COVID-19, the Republic of Korea, which was exposed to COVID-19 early in the pandemic due to its geographic proximity to China, recommended social distancing, using a face mask, frequent handwashing, and refraining from outdoor activities [19]. Furthermore, the country used various media channels to continuously provide information regarding the most recent status of COVID-19 and the latest preventative practices. Free hand sanitizers and face masks were offered to the public for a specific period. Moreover, free disinfection services were offered to facilities visited by COVID-19 patients [19]. Such examples of social support played a pivotal role in encouraging the public to practice infection prevention behaviors [20]. Thus, this study aimed to investigate the relationship between personal hygiene habits, infection prevention behaviors, and the effects of social support on consumers' infection prevention behaviors.

### 1.1. Theoretical Background

Factors Influencing Infection Prevention Behaviors

The social cognitive theory describes the interactions between environmental, individual, and behavioral factors [21]. This theory describes the effects of various individual factors on behavior, one of which is self-efficacy. Self-efficacy is defined as having confidence in one's abilities to organize and execute required behaviors to perform a task [17]. Bandura [18] proposed that accomplishment experience, vicarious experience, verbal persuasion, and physiological state were the antecedents of self-efficacy. Specifically, past accomplishment experience was viewed as the most potent source of self-efficacy [17].

A habit is conceptualized as a learned activity that automatically manifests in a specific situation. Aarts and Dijksterhuis [12] viewed a habit as a "goal-oriented" behavior and stipulated that a habit is formed to attain a certain objective or final state. Verplanken [22] proposed that habits were formed to increase life's vigor. Some studies on habits, such as that by Limayem and Hirt [15], stated that behavior cannot be solely explained by intention and that a force of habit explains a considerable proportion of behaviors. Kim and Yun [23] also reported that habits were predictors of health-promoting behaviors. In this way, we can see that habits and self-efficacy are important predictors of human behavior. In other words, the public's habits are learned behaviors based on their past accomplishments and will influence their future behaviors.

Lally et al. [24] stated that habits are characterized by automaticity and efficiency and that they are learned processes that trigger an impulse for human behavior. In a study on the influence of habits and self-efficacy on learning outcomes, Hamann et al. [25] reported that learning habits and self-efficacy impacted the learners' future behavioral processes. Lee [26] also reported that habits were significantly correlated with self-efficacy. Moreover, in studies on personal habits and self-efficacy in relation to health behaviors, Stuckey et al. [27] and Park [28] reported that health-related habits had positive effects on self-efficacy.

**Hypothesis 1 (H1).** *The public's personal hygiene habits will positively affect their self-efficacy for infection prevention.*

### 1.2. Self-Efficacy for Infection Prevention and Prevention Behaviors

Self-efficacy refers to the belief in one's abilities to organize and perform a particular action to obtain a particular outcome. People with high self-efficacy can deal with challenges better than their counterparts, demonstrating that self-efficacy is an important factor in social adjustment and problem-solving ability [18]. Self-efficacy is divided into action and maintenance self-efficacy. Action self-efficacy refers to the trust in one's abilities to be involved in an action that is yet to be adopted or initiated, while maintenance self-efficacy refers to the trust in one's ability to maintain and continue an action that has already been adopted and initiated [29–31]. In a study on the public's infection prevention behaviors

during the COVID-19 pandemic, Lin et al. [32] reported that action self-efficacy influenced behavior through intention. Di Maio et al. [33] also reported that action self-efficacy affected physical activity planning, while Kim and Yun [23] reported that self-efficacy positively affects health-promoting behaviors.

Several behaviors that can prevent the spread of COVID-19 and other viral infections have been identified. Korea's Central Disaster and Safety Countermeasures Headquarters [19] continuously recommends that the public practice social distancing, wash their hands frequently, wear face masks, and refrain from engaging in activities outside the home as much as possible. Galea et al. [3] stated that social distancing played a key role in halting the spread of COVID-19. Furthermore, Goldberg et al. [34] classified the public's viral prevention behaviors into viral spread-prevention behaviors, such as physical distancing, hand sanitizing, cleaning, maintaining personal hygiene, and purchasing products. Smith et al. [4] reported that the role of purchasing products for infection prevention, such as face masks, antibacterial disinfectants, and sanitizing soaps, was inevitable when considered in terms of preventing COVID-19 infection and that people continued to strive to buy these goods. These results established that public behavior concerning COVID-19 can be divided into the following categories: behaviors related to preventing the spread of the virus and purchasing infection-prevention products.

**Hypothesis 2 (H2).** *The public's self-efficacy for infection prevention will positively affect viral spread-prevention behaviors.*

**Hypothesis 3 (H3).** *The public's self-efficacy for infection prevention will positively affect product-purchasing behaviors for infection prevention.*

*1.3. Infection Prevention and Social Support*

Social support encompasses all positive resources that can be obtained from social relationships and refers to interacting with others or receiving help from others to fulfill social needs [35]. Lee [36] stated that people who were facing danger or disaster interacted with various dimensions of their environments and that social support significantly impacted them. In a study that used an integrated behavior change model to investigate the public's prevention behaviors during the COVID-19 pandemic, Chan et al. [37] also identified the social context that facilitated autonomous motivation as an antecedent for the participants' prevention behaviors, thus highlighting social support as an important factor for individuals in the prevention of COVID-19.

Social support has an indirect positive effect on self-efficacy and individual behaviors [38,39]. Multiple early studies that analyzed the relationship between self-efficacy on health and exercise behaviors [40–42] reported that support from others had a positive impact on individuals' behaviors related to protecting their health. Particularly, Song and Yoo [20] reported that social support had a positive effect on the public's self-efficacy in the COVID 19 setting. Furthermore, Chang et al. [43] utilized social support as a major moderator when classifying peoples' disease-related behaviors. Song [44] defined social support as a positive resource obtained through interactions with others and stated that social support moderates engagement in certain behaviors by boosting one's willingness to engage.

These studies demonstrated that social support was an important moderator of the public's infection-prevention behaviors during infectious disasters, such as the COVID-19 pandemic.

**Hypothesis 4 (H4).** *Social support will moderate the effect of self-efficacy on viral spread-prevention behaviors related to infection prevention.*

**Hypothesis 5 (H5).** *Social support will moderate the effect of self-efficacy on product-purchasing behaviors related to infection prevention.*

*1.4. Conceptual Model*

We established the abovementioned hypotheses and the following study model to examine the relationship between sanitary living habits and self-efficacy in preventing infectious diseases and behaviors during the COVID-19 pandemic. Five hypotheses were developed to examine the relationships between personal hygiene habits, self-efficacy for infection prevention, viral spread-prevention behaviors, product-purchasing behaviors for infection prevention, and social support, as seen in Figure 1.

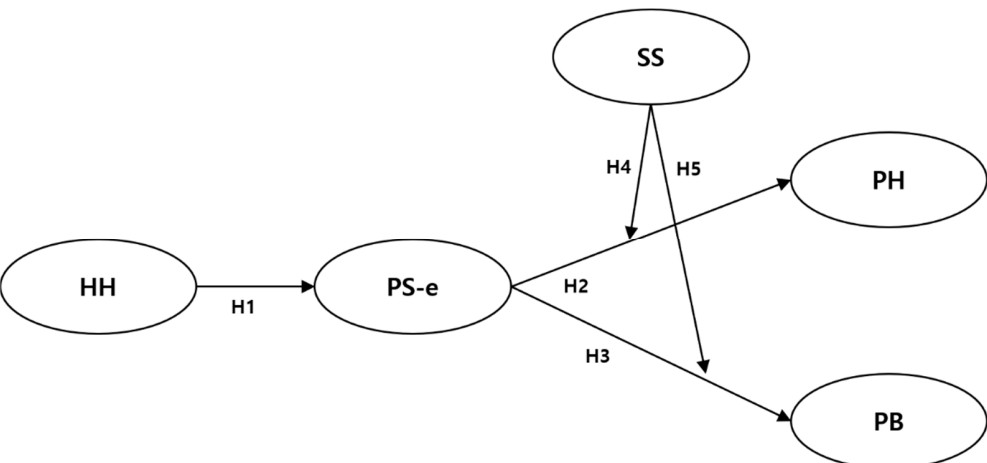

**Figure 1.** The conceptual model. Abbreviations: HH, personal hygiene habits; PS-e, self-efficacy for preventing infectious disease; PH, viral spread-prevention behaviors; PB, product-purchasing behaviors for infection prevention; SS, social support.

## 2. Materials and Methods

*2.1. Data Collection*

We collected data using a self-report questionnaire. The questionnaire was administered to adults for eight days, from 18 to 25 May 2020, by an online survey agency called Macomil Embrain. This agency has a survey panel of 1,249,392 adults, accounting for 3.5% of the South Korean adult population. The questionnaire was revised to a web survey for the convenience of online responses. The web survey was developed such that missing responses or outliers were not allowed. The survey was randomly sent via email to the adult panels of the agency. The questionnaires were retrieved in order depending on the sample allocated based on sex, age, and area of residence. A total of 620 questionnaires were collected, and we checked for any missing responses, outliers, and duplicate responses. We used the data for analysis after confirming that the data had no problems.

*2.2. Measures*

We developed several items (Table 1) to measure the participants' personal hygiene habits, self-efficacy, personal hygiene behaviors for infection prevention, purchasing behaviors for infection prevention, and social support.

### 2.2.1. Personal Hygiene Habits

Personal hygiene habits were measured with reference to the studies that were performed by Lee et al. [45], Rayamajhi et al. [46], Park and Jung [47], Begum et al. [48], and Barbosa et al. [49]. They measured the participants' personal hygiene practices in daily life and at work based on parameters such as handwashing, coughing etiquette, keeping surroundings clean, and managing toiletries. Therefore, our survey items were developed regarding the items that were used in the aforementioned scale. All the items were measured using a five-point Likert scale, ranging from 1 as "strongly disagree" to 5 as "strongly agree".

**Table 1.** Constructs and Survey Questionnaire.

| |
|---|
| Construct 1: Personal hygiene habits(HH) |
| HH1: I wash my hands frequently. |
| HH2: I practice coughing etiquette. |
| HH3: I ventilate rooms often. |
| HH4: I keep my toiletries clean. |
| HH5: I always wash my hands and feet after returning to my home. |
| Hb6: I keep my surroundings clean. |
| Construct 2: Self-efficacy(PS-e) |
| PS-e1: I am well aware of how to protect myself from COVID-19. |
| PS-e2: I can exercise self-control to protect myself from COVID-19. |
| PS-e3: I can try to exercise self-control to protect myself from COVID-19. |
| Construct 3: Viral spread-prevention behaviors(PH) |
| PH1: I practice social distancing to combat the COVID-19. |
| PH2: I avoid going outside and try to stay at home as much as possible to combat the COVID-19. |
| PH3: I always wear a face mask when going outside to combat the COVID-19. |
| PH4: I wash my hands frequently to combat the COVID-19. |
| Construct 4: Product-purchasing behaviors for infection prevention(PB) |
| PB1: I purchased face masks to combat the COVID-19. |
| PB2: I purchased sanitizing soaps to combat the COVID-19. |
| PB3: I purchased antibacterial disinfectants to combat the COVID-19. |
| Construct 5: Social support(SS) |
| SS1: Our society encourages me to engage in actions that assist with combating the COVID-19 crisis. |
| SS2: Our society provides me with infection prevention information that assists with combating the COVID-19 crisis. |
| SS3: Our society provides me with infection prevention products that assist with combating the COVID-19 crisis. |

### 2.2.2. Self-Efficacy for Infection Prevention

Self-efficacy for infection prevention was measured using the scale that Song and Yoo proposed [20]. They developed three items that measured the self-efficacy for infection prevention during the COVID-19 pandemic by modifying the self-efficacy scale created by Floyd et al. [50]. Each item was rated using a five-point Likert scale (range, 1 as "strongly disagree" to 5 as "strongly agree").

### 2.2.3. Infection-Prevention Behaviors

The public's infection-prevention behaviors in response to the COVID-19 pandemic were divided into the viral spread-prevention behaviors and product-purchasing behaviors for infection prevention regarding the studies that were performed by Goldberg et al. [34]; Galea et al. [3]; Korea's Central Disaster and Safety Countermeasures Headquarters, Central Disease Control Headquarters [19]; and Smith et al. [4]. In particular, Goldberg et al. [34] developed the following survey items that assessed the behaviors that were related to the prevention of the spread of the virus during the COVID-19 pandemic: "kept at least six feet away from [other people] outside of [the] home", "[avoided] parties and other personal events", "washed [their] hands with soap and water [more frequently]", and "[wore a] mask in public to protect [themselves] or others". Further, Goldberg et al. [34] and Smith et al. [4] categorized product-purchasing behaviors (e.g., bought protective masks) as infection prevention behaviors. Thus, in this study, we developed four and three survey items that addressed the viral spread-prevention behaviors and the product-purchasing behaviors for infection prevention, respectively. Each item was rated using a five-point Likert scale (range, 1 as "strongly disagree" to 5 as "strongly agree").

### 2.2.4. Social Support

Social support was assessed using the emotional, infection-prevention products, and information support scales that were adapted for use in the COVID-19 context in Korea by Song and Yoo [20] and Song [44]. Each item was rated using a five-point Likert scale (range, 1 as "strongly disagree" to 5 as "strongly agree").

*2.3. Analysis*

All statistical analyses were performed using SPSS software version 26.0 and SPSS AMOS software version 21.0. The respondents' demographic characteristics were analyzed using frequency analysis. The baseline values were presented as the mean (M) and standard deviation (SD). The reliability of the scale was evaluated using confirmatory factor analysis (CFA) and reliability analysis. The model fit in the CFA was determined with reference to the criteria that were proposed by Joreskog and Sorbom [51], Byrne [52], and Tobbin [53]. Furthermore, the scale's reliability was evaluated with the composite reliability (CR), average variance extracted (AVE), and square roots of the AVEs using the standardized estimate and variance estimate obtained in the CFA. The Cronbach's $\alpha$ that was obtained from the results was compared with those reported by Fornell and Larchker [54], Nunnally [55], and Chen et al. [56] H1–H3 were tested using structural equation modeling (SEM). H4–H5 were tested with reference to the moderation analysis method that was proposed by Woo [57]. First, the mean social support scores were used to classify the participants according to specific cutoff values (≥mean and <mean). Next, multi-group SEM was conducted by constraining each path, and the standardized path coefficients between two corresponding groups in an unconstrained model were compared. A statistically significant difference between the path coefficients was recognized if the absolute value of the composite reliability between the standardized path coefficients for each group was ≥1.965 ($p > 0.05$). The variable was deemed to have a moderating effect if this requirement was met.

# 3. Results

*3.1. Samples*

A total of 620 cases were included from the data in the final analysis. The respondents' demographic characteristics are shown in Table 2. Of the total respondents, 51.8% were male and 48.2% were female. The ages included respondents who were in their 20 s, 30 s, 40 s, 50 s, and ≥60 s (18.1%, 17.6%, 21.5%, 23.5%, and 19.4%, respectively). The education levels included high school graduates or lower, associate degrees, bachelor's degrees, and master's degrees or higher (20.0%, 16.1%, 55.3%, and 8.5%, respectively). The occupations included full-time workers, hourly workers, self-employed persons, housewives, students, and other or unemployed (49.4%, 5.5%, 11.6%, 13.9%, 7.1%, and 12.6%, respectively). The average monthly household income was 4.78 (±3.17 million) KRW. This study was approved by the Research Ethics Committee of Chungbuk National University (CBNU 202006-0109).

*3.2. Measurement Model*

To determine the suitability of the measurement model, we assessed its content, convergent, and discriminant validity. First, the content validity was tested by developing constructs and measurement items based on the previous studies. Second, the convergent validity was tested by examining the fit indices that were computed in the CFA. The results demonstrated that PH2, one of the items for viral spread-prevention behaviors, decreased the scale's reliability, therefore, this item was removed. The CFA was repeated, and the results were as follows: chi-square distribution ($\chi^2$/df) = 2.884, root mean square residual (RMR) = 0.037, root mean square error of approximation (RMSEA) = 0.055, goodness of fit index (GFI) = 0.938, adjusted GFI (AGFI) = 0.914, normed fit index (NFI) = 0.933, relative fit index (RFI) = 0.916, incremental fit index (IFI) = 0.955, Tucker Lewis index (TLI) = 0.944, and comparative fit index (CFI = 0.955). As a result, a good model fit was confirmed. Additionally, we assessed the standardized estimate, Cronbach's $\alpha$, CR, and AVE. The standardized estimate was over o.6 points, Cronbach's $\alpha$ and CRs for each construct were over 0.7 points, and the AVE for each item was higher than 0.5 points (Table 3) [54]. Therefore, the convergent validity was supported. The discriminant validity was tested using the inter-construct correlation coefficients. The square roots of the AVEs for each construct marked in bold in Table 3 are higher than those for the other values [55].

**Table 2.** Demographic characteristics for the samples.

| Characteristics | Frequency | Percentage |
|---|---|---|
| Sex | | |
| Male | 321 | 51.8 |
| Female | 299 | 48.2 |
| Age | | |
| 20–29 | 112 | 18.1 |
| 30–39 | 109 | 17.6 |
| 40–49 | 133 | 21.5 |
| 50–59 | 146 | 23.5 |
| ≥60 | 120 | 19.4 |
| Residence | | |
| Metropolitan area | 264 | 42.6 |
| Non-metropolitan area | 356 | 57.4 |
| Education | | |
| High school or lower | 124 | 20.0 |
| Vocational school | 100 | 16.1 |
| Bachelor's degree | 343 | 55.3 |
| Master's degree or higher | 53 | 8.5 |
| Occupation | | |
| Full-time employee | 306 | 49.4 |
| Part-time employee | 34 | 5.5 |
| Self-employed | 72 | 11.6 |
| Student | 44 | 7.1 |
| Housewife | 86 | 13.9 |
| Unemployed or other | 78 | 12.6 |
| Monthly household income * | M = 4.78 million KRW | SD = 3.17 million KRW |

Note: * 10,000 South Korean won (USD 1 = KRW 1117.600). Abbreviations: M, mean; SD, standard deviation.

**Table 3.** The standardized estimates, cronbach's $\alpha$, CR, AVE, inter-construct correlations, and means for the study variables.

| Latent Variable | Measurement Variable | Standar—Dized Estimate | Cronbach's $\alpha$ | CR | AVE | Interconstruct Correlations | | | | | Mean (SD) |
|---|---|---|---|---|---|---|---|---|---|---|---|
| | | | | | | HH | PS-e | PH | PB | SS | |
| Personal hygiene habits | HH1 | 0.608 | 0.812 | 0.844 | 0.519 | 0.720 | | | | | 243.824 (0.644) |
| | HH2 | 0.775 | | | | | | | | | |
| | HH3 | 0.696 | | | | | | | | | |
| | HH4 | 0.731 | | | | | | | | | |
| | HH5 | 0.730 | | | | | | | | | |
| | HH6 | 0.774 | | | | | | | | | |
| Self-efficacy for disease prevention | PS-e1 | 0.863 | 0.906 | 0.912 | 0.776 | 0.116 | 0.881 | | | | 4.082 (0.773) |
| | PS-e2 | 0.876 | | | | | | | | | |
| | PS-e3 | 0.801 | | | | | | | | | |
| Viral spread-prevention behavior | PH1 | 0.678 | 0.827 | 0.900 | 0.752 | 0.344 | 0.511 | 0.867 | | | 4.317 (0.636) |
| | PH3 | 0.818 | | | | | | | | | |
| | PH4 | 0.860 | | | | | | | | | |
| Product-purchasing behaviors for infection prevention | PB1 | 0.731 | 0.802 | 0.792 | 0.561 | 0.352 | 0.141 | 0.393 | 0.749 | | 3.553 (0.887) |
| | PB2 | 0.866 | | | | | | | | | |
| | PB3 | 0.857 | | | | | | | | | |
| Social support | SS1 | 0.678 | 0.808 | 0.840 | 0.638 | 0.137 | 0.399 | 0.251 | 0.161 | 0.799 | 3.643 (0.790) |
| | SS2 | 0.768 | | | | | | | | | |
| | SS3 | 0.894 | | | | | | | | | |

### 3.3. Structural Model 1

The structural model was determined to have an acceptable fit based on the following fit indices: $\chi^2/df$ = 3.368, RMR = 0.068, RMSEA = 0.052, GFI = 0.945, AGF = 0.921, NFI = 0.936, IFI = 0.954, and CFI = 0.954. Figure 2 illustrates the structural model.

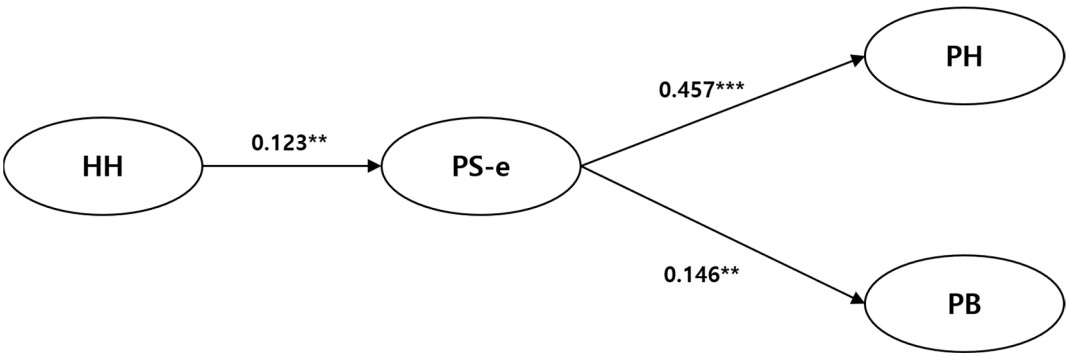

**Figure 2.** Structural model with the standardized path estimate. Note: ** $p < 0.01$, *** $p < 0.001$. Abbreviations: HH, personal hygiene habits; PS-e, self-efficacy for preventing infectious disease; PH, virus spread prevention behaviors; PB, product-purchasing behaviors for infection prevention.

First, personal hygiene habits positively affected self-efficacy for infection prevention ($\beta$ = 0.123, $p < 0.01$), thus supporting H1. Second, self-efficacy for infection prevention had a statistically significant effect on the viral spread-prevention behaviors and product-purchasing behaviors for infection prevention ($\beta$ = 0.457 and 0.146; $p < 0.000$ and $p < 0.01$, respectively), thus supporting H2 and H3.

### 3.4. Structural Model 2

The participants were divided into two groups based on whether their social support score was >3 or ≤3. A total of 392 participants had the perception that they received high psychological support, while 228 had the perception that they received low psychological support (M = 4.314 and 2.768, SD = 0.465 and 0.0337, respectively). A total of 476 participants had the perception that they received high informational support, while 144 had the perception that they received low informational support (M = 4.305 and 2.764, SD = 0.461 and 0.487). A total of 269 participants had the perception that they received high material support, while 351 had the perception that they received low material support (M = 4.260 and 2.450, SD = 0.269 and 0.691).

Table 4 demonstrates the differences in the self-efficacy for infection prevention and the level of infection prevention behaviors between the groups. Participants who had the perception that they received high social support had mean scores for all the significantly higher variables than those of their counterparts.

**Table 4.** Differences in the means for the self-efficacy for infection prevention and infection prevention behaviors according to the perceived social support.

| Variable | Group | N | Self-Efficacy for Infection Prevention | | | Viral Spread-Prevention Behaviors | | | Product-Purchasing Behaviors for Infection Prevention | | |
|---|---|---|---|---|---|---|---|---|---|---|---|
| | | | M | SD | *t*-Value | M | SD | *t*-Value | M | SD | *t*-Value |
| Psychological support | Low | 228 | 3.829 | 0.736 | −6.408 *** | 4.192 | 0.666 | −3.700 *** | 3.408 | 1.074 | −2.463 * |
| | High | 392 | 4.229 | 0.757 | | 4.390 | 0.608 | | 3.637 | 1.140 | |
| Informational support | Low | 144 | 3.711 | 0.821 | −6.811 *** | 4.093 | 0.713 | −4.923 *** | 3.387 | 1.071 | −2.035 * |
| | High | 476 | 4.194 | 0.722 | | 4.385 | 0.596 | | 3.603 | 1.132 | |
| Material support | Low | 351 | 3.954 | 0.742 | −4.766 *** | 4.264 | 0.641 | −2.386 * | 3.406 | 1.099 | −3.750 *** |
| | High | 269 | 4.248 | 0.782 | | 4.387 | 0.624 | | 3.743 | 1.122 | |

Note: * $p < 0.05$, *** $p < 0.001$. Abbreviations: M, Mean; SD, standard deviation.

A multi-group SEM was performed to evaluate the moderating effects of each social support component. The results confirmed that informational support and material support had moderating effects. First, Figure 3 shows the moderating effects of informational support. Informational support strengthened the effect of self-efficacy for infection prevention on viral spread-prevention behaviors (model fit: $\chi^2/\mathrm{df} = 2.682$, RMR = 0.650, RMSEA = 0.052, GFI = 0.914, AGFI = 0.879, CFI = 0.932). Second, Figure 4 shows the moderating effects of material support. Providing products for infection prevention diminished both the viral spread-prevention behaviors and product-purchasing behaviors for infection prevention (model fit: $\chi^2/\mathrm{df} = 2.813$, RMR = 0.078, RMSEA = 0.054, GFI = 0.911, AGFI = 0.874, CFI = 0.928). Contrastingly, psychological support did not have a moderating effect.

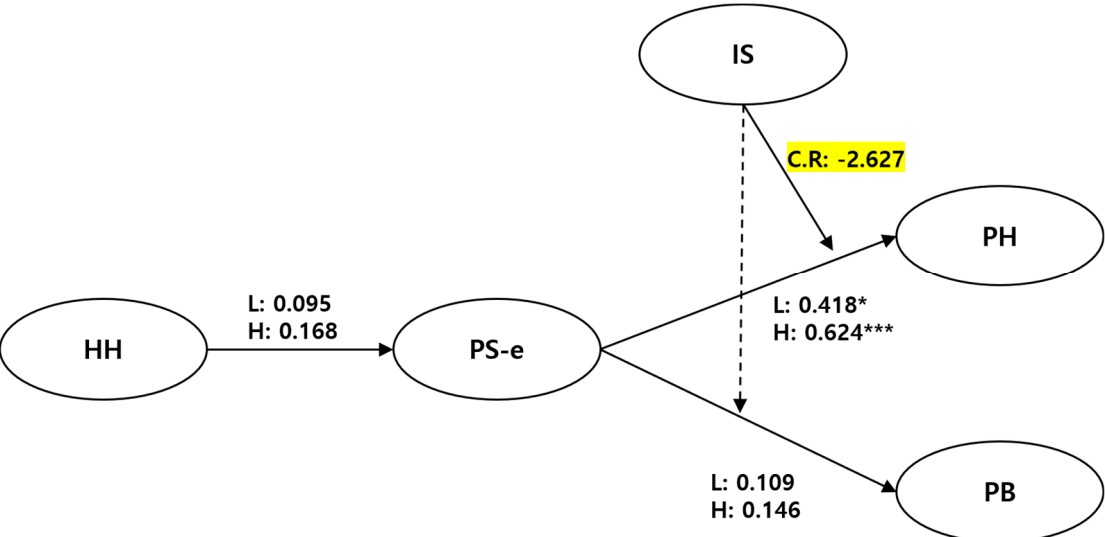

**Figure 3.** Structural model comparing the standardized path estimate and composite reliability in the high and low informational support groups. Note: * $p < 0.05$, *** $p < 0.001$. Abbreviations: HH, personal hygiene habits; PS-e, self-efficacy for preventing infectious diseases; PH, viral spread-prevention behaviors; PB, product-purchasing behaviors for infection prevention; IS, information support.

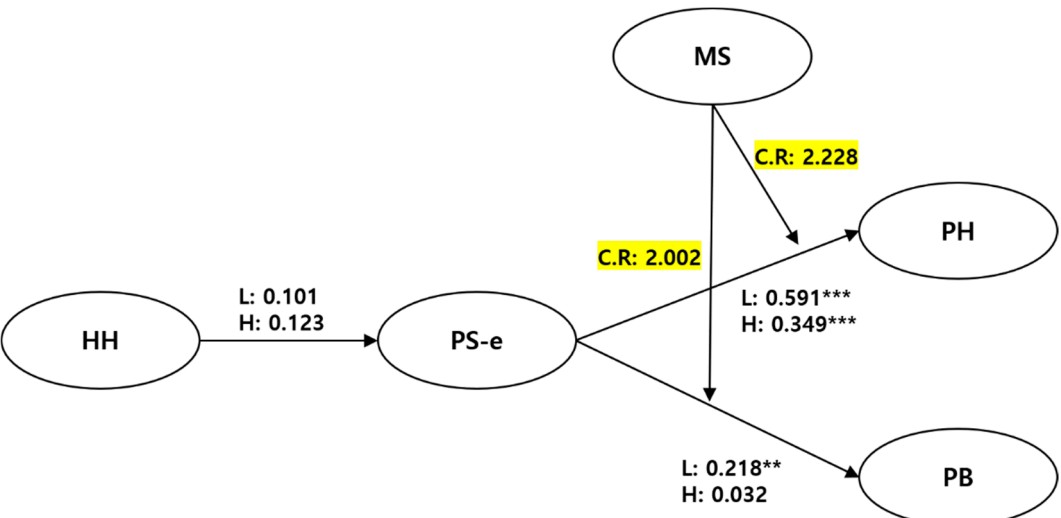

**Figure 4.** Structural model comparing the standardized path estimate and composite reliability in the high and low product support for infection prevention groups. Note: ** $p < 0.01$, *** $p < 0.001$. Abbreviations: HH, personal hygiene habits; PS-e, self-efficacy for preventing infectious diseases; PH, viral spread-prevention behaviors; PB, product-purchasing behaviors for infection prevention; MS, material support.

## 4. Discussion

This study investigated the relationship between personal hygiene habits, consumers' infection-prevention behaviors, and the effect of social support on consumers' infection-prevention behaviors. We aimed to determine the type of support that should be provided to the public to effectively prevent the spread of infection during an outbreak.

The results were as follows. First, personal hygiene habits had a positive effect on self-efficacy for infection prevention and had an indirect positive effect on infection prevention behaviors through self-efficacy for infection prevention. This highlights the importance of complying with personal hygiene practices in daily life, regardless of whether there is an outbreak happening at the time. This is supported by the studies by Kim and Yun [23], Stuckey et al. [27], Kerstin et al. [25] that showed that health-related habits affect health promotion behavior. Second, the public demonstrated a higher self-efficacy for infection prevention and compliance with infection-prevention behaviors when combined with higher perceived social support for infection prevention. Furthermore, informational support increased the effect of the public's self-efficacy for infection prevention on viral spread-prevention behaviors. These findings are supported by the findings of Yoo and Joo [58], Kim et al. [59] that information support for risk factors affects individuals' risk response actions. This result confirms the importance of informational support. Thus, we propose that it is important to provide the public with quality information during an infectious disaster. Furthermore, measures should be taken to ensure that the public feels that they have been provided with appropriate information to prevent and respond to the pandemic. This notion was supported by the results from the study by Ali and Bhatti [60] that stated that public health information should be imparted to the public through various channels during the COVID-19 pandemic. Moreover, Wallace et al. [61] demonstrated that Canadians responded to the provision of public health information during the COVID-19 pandemic. In contrast, material support lowered the public's compliance with infection-prevention behaviors. This suggests that an excess supply of infection prevention products may cause the public to relax their infection prevention efforts. Thus, an environment that enables people to sanitize their hands at any time can result in less rigorous hand sanitizing practices by the public. This is similar to the concept related to insensitivity toward safety [62].

This study has several limitations. First, several factors could have affected the public's infection prevention behaviors during a pandemic; however, this study only focused on personal hygiene habits and social support. In the future, research that considers policy agreements to prevent infectious disease and people's levels of depression should be considered. Second, the study population only comprised Koreans. Infections, such as COVID-19, spread across national borders and are particularly serious in countries with high population densities. Thus, subsequent studies should be expanded to include a greater number of countries than this study. Third, sample recruitment was performed using an online survey. Thus, the study sample only comprised people who used the internet. Fourth, this study did not take demographic factors, such as sex and socioeconomic position, into consideration. Subsequent studies should consider the demographic characteristics of the participants. Lastly, all variables, including personal hygiene habits, were assessed through a self-reported questionnaire that increased the risk of self-reporting bias.

## 5. Conclusions

Based on the results, we suggest that the following safety and management measures be taken to promote the public's practice of preventative behaviors during outbreaks. First, the public should receive education around practicing daily personal hygiene habits. From children and adolescents through to childcare centers or school programs and, finally, to adults, personal hygiene practices should be reinforced through the continuous sharing of information through social education or campaigns. Second, during outbreaks, infection prevention practices and information should be shared through various media in a timely manner to allow for people to comply with these practices. However, depending on the cul-

ture and context, the interpretations of the same piece of information can vary extensively. Information should be processed and disseminated with consideration of the context of the information recipients. Third, material support reduces the effect of consumers' self-efficacy for infection prevention on personal hygiene behaviors and product-purchasing behaviors. Thus, because excessive material support can increase the consumers' reliance on society, the provision of infection prevention products should be dependent on the citizens' social welfare status.

**Author Contributions:** Conceptualization, H.J.Y. and E.S.; methodology, E.S.; software, E.S.; validation, H.J.Y.; formal analysis, E.S.; investigation, E.S.; resources, H.J.Y.; data curation, E.S.; writing—original draft, H.J.Y. and E.S.; writing—review and editing, E.S. and H.J.Y.; visualization, E.S.; supervision, H.J.Y. All authors have read and agreed to the published version of the manuscript.

**Funding:** This research was supported by Chungbuk National University Korea National University Development Project (2020).

**Institutional Review Board Statement:** This study was approved by the Research Ethics Committee of Chungbuk National University (CBNU 202006-0109) and was carried out following the rules of the Declaration of Helsinki of 1975. Informed consent was obtained from the participants.

**Conflicts of Interest:** The authors declare no conflict of interest.

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
