# Peer review of "Effects of Personal Hygiene Habits on Self-Efficacy for Preventing Infection, Infection-Preventing Hygiene Behaviors, and Product-Purchasing Behaviors"

_sustainability, doi:10.3390/su13179483_

Round 1
Reviewer 1 Report
The study seems well conducted. I have some questions.
- Did the pattern of results differ between individuals that already experienced prior epidemics (when they were at least adolescents / better 18 years old) compared to those who did not? That means, are experienced people more likely to show an adapted behavior?
- Did the pattern of associations between variables vary by sex, socioeconomic position, etc.? That means, are the patterns universal across individuals or specific for certain subpopulations?
- Are the results specific for the current COVID-19 pandemic or may they also be generalizable to other (future) pandemics? The applied implications of this issue seem of importance.
Author Response
□ Comment 1
|
Reviewers’ comment: Did the pattern of results differ between individuals that already experienced prior epidemics (when they were at least adolescents / better 18 years old) compared to those who did not? That means, are experienced people more likely to show an adapted behavior? |
Response: Thank you for reading our manuscript and providing your valuable comments. We have responded to your questions below. This study was only conducted on adults. Thus, our study does not shed light on the contrast between individuals under 18 and adults aged 18 years or older. We stated in the manuscript that this study is only conducted on adults aged 18 years or older as below:
□ Comment 2
|
Reviewers’ comment: Did the pattern of associations between variables vary by sex, socioeconomic position, etc.? That means, are the patterns universal across individuals or specific for certain subpopulations? |
Response: In this study, we did not consider sex and socioeconomic position. We only examined the overall individual tendencies. Thus, we presented this as a limitation as below(please see page 14, lines 442-445):
‘Fourth, this study did not take demographic factors, such as sex and socioeconomic position, into consideration. Subsequent studies should consider the demographic characteristics of the participants.’
□ Comment3
|
Reviewers’ comment : Are the results specific for the current COVID-19 pandemic or may they also be generalizable to other (future) pandemics? The applied implications of this issue seem of importance. |
Response: The results are specific to the COVID-19 pandemic. However, as COVID-19 is an unprecedented infectious pandemic, the results of this study could be used as a reference in future pandemics.
++We appreciate the insightful comments of the reviewers,
which have helped us improve the quality of our manuscript. Thank you! ++
Reviewer 2 Report
The authors investigated the relationship between personal hygiene habits, infection prevention behaviors, and the effects of social support on consumers’ infection prevention behaviors.
The main results showed that personal hygiene habits: 1) had a positive effect on self-efficacy for infection prevention; 2) indirectly affected virus spread prevention behaviors; 3) product purchasing behaviors for infection prevention through self-efficacy for infection prevention.
The research methodology is suitable. I suggest major revisions before the acceptation.
The manuscript is of general interest. I have the following major comments.
ABSTRACT
-The background is not appropriate for the purpose of the study. It would be more appropriate to emphasize the health recommendations and hygiene measures adopted during COVID-19 (for example, frequent handwashing), as reported in the introduction of the main text.
-Participant information should be more detailed (gender distribution, age range)
-Authors should specify the type of tool used and what characteristics it investigated
METHODS
-Methods should be repeatable. Therefore, please specify the online platform used. Furthermore, it is not clear how the sample was recruited. The authors stated that they distributed the questionnaire by mail. Were the recipients of the emails part of a specific mailing list?
-The authors stated that they collected 620 questionnaires. All completed questionnaires were valid or some questionnaires were removed in the cleaning process? For example, for multiple submissions of the same respondent or for lack of all responses within the questionnaire
DISCUSSION
-The discussion does not represent a considerable extension of the results. Authors should argue the results found by referring to the scientific literature in agreement and in contrast.
-Among the limitations of the study, authors should report that personal hygiene habits were assessed through a self-reported questionnaire increasing the risk of self-reporting bias.
Author Response
□ Comment 1
|
Reviewers’ comment: (Abstract) 1. The background is not appropriate for the purpose of the study. It would be more appropriate to emphasize the health recommendations and hygiene measures adopted during COVID-19 (for example, frequent handwashing), as reported in the introduction of the main text. 2. Participant information should be more detailed (gender distribution, age range) 3. Authors should specify the type of tool used and what characteristics it investigated. |
Response: We appreciate your reviews and agree with the comments. In response to the reviewers' comments, the ABSTRACT has been revised as follows. However, participant information is briefly presented by referring to the regulations and other published articles in this journal(Sustainability)(please see page 1, lines 10-17):
‘Since there is no cure for the COVID-19 pandemic yet, personal hygiene management is important for protecting oneself from the deadly virus. Personal hygiene management comes from personal hygiene habits. Thus, this study investigated the association between personal hygiene habits, consumers’ infection-prevention behaviors, and the effects of social support on the latter. Data were collected using a self-administered questionnaire survey from 620 Korean adults. An online survey agency was used to conduct the questionnaire over 8 days, from May 18 to May 25, 2020. Data were analyzed using structural equation modeling.’
□ Comment 2
|
Reviewers’ comment : (Method) 1. Methods should be repeatable. Therefore, please specify the online platform used. Furthermore, it is not clear how the sample was recruited. The authors stated that they distributed the questionnaire by mail. Were the recipients of the emails part of a specific mailing list? 2. The authors stated that they collected 620 questionnaires. All completed questionnaires were valid or some questionnaires were removed in the cleaning process? For example, for multiple submissions of the same respondent or for lack of all responses within the questionnaire |
Response: Yes, you are right. We fully agree with you. The 3.1 Data Collection section in the METHOD has been revised as follows (please see page 5, lines 194-205):
‘We collected data using a self-report questionnaire. The questionnaire was administered to adults for 8 days, from May 18 to 25, 2020, by an online survey agency called Macomil Embrain. This agency has a survey panel of 1,249,392 adults, accounting for 3.5% of the Korean adult population. The questionnaire was revised to a web survey for the convenience of online responses. The web survey was developed such that missing responses or outliers were not allowed. The survey was randomly sent via email to the adult panels of the agency. The questionnaires were retrieved in order depending on the sample allocated based on sex, age, and area of residence. A total of 620 questionnaires were collected, and we checked for any missing responses, outliers, and duplicate responses. We used the data for analysis after confirming that the data had no problems.’
□ Comment 3
|
Reviewers’ comment : (Discussion) The discussion does not represent a considerable extension of the results. Authors should argue the results found by referring to the scientific literature in agreement and in contrast. |
Response: Yes, you are right. The DISCUSSION section has been revised as follows (please see page 13, lines 402-416):
'The results were as follows. First, personal hygiene habits had a positive effect on self-efficacy for infection prevention and had an indirect positive effect on infection prevention behaviors through self-efficacy for infection prevention. This highlights the importance of complying with personal hygiene practices in daily life, regardless of whether there is an outbreak happening at the time. This is supported by the studies by Kim and Yun[23], Stuckey et al.[27], Kerstin et al.[58] that health-related habits affect health promotion behavior. Second, the public demonstrated a higher self-efficacy for infection prevention and compliance with infection-prevention behaviors when combined with higher perceived social support for infection prevention. Furthermore, informational support increased the effect of the public’s self-efficacy for infection prevention on viral spread-prevention behaviors. These findings are supported by the findings of Yoo and Joo[59], Kim et al.[60] that information support for risk factors affects individuals’ risk response actions. This result confirms the importance of informational support.'
++ references
- Kim, M.; Yun, S. Effects of eating habits and self-efficacy on nursing students' health promotion behaviors: In convergence era. Soc. for SMB2017, 7, 111–117. DOI:10.22156/cs4smb.2017.7.2.111
- Stuckey, M.; Shapiro, S.; Gill, D.; Petrella, R. A lifestyle intervention supported by mobile health technologies to improve the cardiometabolic risk profile of individuals at risk for cardiovascular disease and type 2 diabetes: study rationale and protocol. BMC Public Health 2013, 13, 1051. DOI:10.1186/1471-2458-13-1051
- Kerstin, H.; Pilotti, M.A.E.; Wilson, B.M. Students’ self-efficacy, causal attribution habits and test grades, Sci. 2020, 10, 231. DOI:10.3390/educsci10090231
- Yoo, H.J.; Joo, S.H. Structural Equation analysis on consumers’ perceived food safety and food safety orientation behavior. Consum. Policy Educ. Rev. 2012, 8, 49–70. 31. DOI:
- Kim, K.H.; Yoo, H.J.; Song, E. An analysis on the structural model for consumer trust-anxiety-competency by source of information-focused on chemical household products. Crisisonomy 2017, 13, 141–158. DOI:10.14251/crisisonomy.2017.13.3.141
□ Comment 4
|
Reviewers’ comment : (Limitation) Among the limitations of the study, authors should report that personal hygiene habits were assessed through a self-reported questionnaire increasing the risk of self-reporting bias. |
Response: Yes, you are right. The following sentence has been added to the LIMITATION section (please see page 14, lines 445-447):
‘Lastly, all variables, including personal hygiene habits, were assessed through a self-reported questionnaire that increased the risk of self-reporting bias.’
++We appreciate the constuctive comments of the reviewers,
which have helped us improve the quality of our manuscript. Thank you! ++

Round 2
Reviewer 2 Report
Dear Editors,
in my opinion the manuscript was improved according to Reviewers' indications. For these reasons it can be accepted for pubblication in the current form.
Best Regards
Giuseppe Battaglia